# Enriched Pathways of Calcium Regulation, Cellular/Oxidative Stress, Inflammation, and Cell Proliferation Characterize Gluteal Muscle of Standardbred Horses between Episodes of Recurrent Exertional Rhabdomyolysis

**DOI:** 10.3390/genes13101853

**Published:** 2022-10-14

**Authors:** Stephanie J. Valberg, Deborah Velez-Irizarry, Zoë J. Williams, Marisa L. Henry, Hailey Iglewski, Keely Herrick, Clara Fenger

**Affiliations:** 1Mary Anne McPhail Equine Performance Center, Department of Large Animal Clinical Sciences, College of Veterinary Medicine, Michigan State University, East Lansing, MI 48824, USA; 2Equine Integrated Medicine, PLC, Lexington, KY 40324, USA

**Keywords:** muscle disease, proteomics, transcriptomics, tying up, myopathy

## Abstract

Certain Standardbred racehorses develop recurrent exertional rhabdomyolysis (RER-STD) for unknown reasons. We compared gluteal muscle histopathology and gene/protein expression between Standardbreds with a history of, but not currently experiencing rhabdomyolysis (*N* = 9), and race-trained controls (*N* = 7). Eight RER-STD had a few mature fibers with small internalized myonuclei, one out of nine had histologic evidence of regeneration and zero out of nine degeneration. However, RER-STD versus controls had 791/13,531 differentially expressed genes (DEG). The top three gene ontology (GO) enriched pathways for upregulated DEG (*N* = 433) were inflammation/immune response (62 GO terms), cell proliferation (31 GO terms), and hypoxia/oxidative stress (31 GO terms). Calcium ion regulation (39 GO terms), purine nucleotide metabolism (32 GO terms), and electron transport (29 GO terms) were the top three enriched GO pathways for down-regulated DEG (*N* = 305). DEG regulated RYR1 and sarcoplasmic reticulum calcium stores. Differentially expressed proteins (DEP ↑*N* = 50, ↓*N* = 12) involved the sarcomere (24% of DEP), electron transport (23%), metabolism (20%), inflammation (6%), cell/oxidative stress (7%), and other (17%). DEP included ↑superoxide dismutase, ↑catalase, and DEP/DEG included several cysteine-based antioxidants. In conclusion, gluteal muscle of RER-susceptible Standardbreds is characterized by perturbation of pathways for calcium regulation, cellular/oxidative stress, inflammation, and cellular regeneration weeks after an episode of rhabdomyolysis that could represent therapeutic targets.

## 1. Introduction

Recurrent exertional rhabdomyolysis (RER) occurs in 6% of Standardbred racehorses, and approximately 7% of Thoroughbred racehorses—particularly young nervous fillies [1,2,3]. The expression of RER is impacted by a genetic predisposition coupled with environmental factors that complicate the identification of susceptibility loci in both breeds [4,5,6]. A lack of a defined genetic predisposition could be due to a polygenic mode of inheritance with small effect size for RER susceptibility, allele specific expression, transcriptional or translational modifications, alternate splicing, or selective isoform expression induced in a race training environment [4,7]. 

In Standardbred horses with RER, muscle histology, biochemistry, exercise responses, electron transport chain coupling, and microarray analysis of gene expression have been investigated during an episode of acute rhabdomyolysis [1,8,9,10,11,12,13]. These studies show that episodes of rhabdomyolysis occur intermittently, characterized by sudden increases in serum creatine kinase activity (CK) without accompanying lactic acidosis [11,12]. Differential expression of Ca^2+^ regulatory, mitochondrial and metabolic genes [9], and altered mitochondrial coupling [13] have been identified during acute rhabdomyolysis in Standardbred horses. Both an abnormality in intramuscular calcium regulation and disturbed mitochondrial function have been proposed as a cause of RER in this breed [9,14,15]. Between episodes of rhabdomyolysis, Standardbred horses are as successful, or more successful racehorses, than horses with no history of RER and it is unknown if altered gene or protein expression persists in the muscle of Standardbred horses at this time point [1,9]. Elucidating characteristic skeletal muscle molecular profiles between episodes of rhabdomyolysis using a multi-omic approach could further define the pathophysiology of RER in Standardbred horses and identify putative preventative therapeutic targets. 

RNA-seq transcriptomics combined with Tandem Mass Tag liquid chromatography mass spectrometry (TMT LC/MS/MS) quantitative proteomics can be used to identify genes and proteins that are differentially expressed in affected horses at greater or lesser levels than control horses. Gene Ontology (GO) enrichment can then be used to identify biological or molecular pathways, and cellular locations associated with differentially expressed genes and proteins [16]. The differences in gene and protein expression with pathway enrichment provide evidence of cellular aberrations that may contribute to the etiology or progression of disease [17,18]. This type of transcriptomic and proteomic analysis was recently performed in Thoroughbred racehorses susceptible to, but not experiencing clinical signs of rhabdomyolysis. Results of that study identified increased protein and decreased gene expression in pathways of calcium regulation and mitochondrial function between episodes of rhabdomyolysis [7]. An underlying cause for RER was hypothesized to be environmental stressors causing post-translational modification of the calcium release channel (RYR1) that triggered excessive release of sarcoplasmic reticulum calcium stores, muscle contracture, mitochondrial calcium buffering and damage [7]. 

We hypothesized that a combined transcriptomic and proteomic approach utilizing Standardbred mares with a history of RER and muscle samples obtained between episodes of rhabdomyolysis would identify chronic underlying characteristics of muscle from RER-susceptible Standardbred horses. Our first objective was to identify differentially expressed gene transcripts (DEG) and their associated biological processes in the muscle of RER-susceptible versus control horses. The second objective was to identify differentially expressed proteins (DEP) and their associated pathways in the muscle of RER-susceptible versus control horses. The third objective was to perform GO enrichment analysis with both DEP and DEG to identify aberrant pathway alterations that exist in the muscle of RER-susceptible Standardbreds between episodes of rhabdomyolysis that could represent preventative therapeutic targets. Lastly, based on the “omic” results, the final objective was to investigate differences in the concentrations of reactive oxygen species (ROS) and the antioxidants glutathione and Coenzyme Q_10_ (CoQ_10_). 

## 2. Materials and Methods

### 2.1. RER-Susceptible Horses

The study included nine RER-susceptible and six control Standardbred mares in full race training, housed at or in the vicinity of The Meadows Racetrack in Pennsylvania USA (Appendix A). RER-susceptible mares had at least two previous episodes of exertional rhabdomyolysis confirmed by a veterinarian through analysis of CK and aspartate aminotransferase (AST) activity. The last reported clinical episode was 6.0 ± 2.9 weeks (range 1–10 weeks) previously. Histories, plasma CK, AST activities, and muscle histopathology were assessed in control horses at the time of sampling to ensure that there was no evidence of rhabdomyolysis. Diets were at the trainer’s discretion and varied between individual horses; however, each concentrate fed to control horses was also fed to at least one RER-susceptible horse (Appendix A). 

### 2.2. Sample Collection

Jugular venous blood samples and muscle biopsies were obtained between one and a half to three hours after exercise to establish CK and AST activities at the time of sampling (Appendix A). The type of exercise performed at The Meadows Racetrack during sampling was determined by each’s horse individual training regime. Four of the RER-susceptible horses were biopsied after racing a mile, four after daily trotting exercise, and one after an hour on a walker. Five control horses were biopsied after racing a mile, one after trotting exercise, and one after walking for an hour. Samples were obtained when horses had no clinical evidence of muscle pain or stiffness to evaluate the chronic underlying basis for RER rather than nonspecific alterations that occur with acute muscle damage. Blood samples were kept on ice packs for up to six hours before they were centrifuged, and plasma frozen prior to analysis of CK and AST at the Michigan State University Veterinary Diagnostic Laboratory. 

For muscle sampling, horses were sedated with 200 mg xylazine (AnaSed, Santa Cruz Animal Health, Santa Cruz, CA, USA) intravenously, and percutaneous gluteus medius muscle biopsies obtained at a standardized site as previously described [19]. Approximately 200 mg of skeletal muscle was divided into two aliquots. One aliquot for proteomic and transcriptomic analyses was immediately frozen in liquid nitrogen and the other, for histochemistry, was rolled in talc before freezing in liquid nitrogen to prevent freeze-artifact formation. Muscle was stored at −80 °C until analysis. 

This study was carried out in strict accordance with the recommendations in the Guide for the Care and Use of Laboratory Animals of the National Institutes of Health. The protocol was approved by the Animal Use and Care Committee of Michigan State University (Proto201900038). 

### 2.3. Muscle Histochemistry

Sections of muscle, measuring 8 µm in thickness, were stained with hematoxylin and eosin (HE), modified Gomori Trichrome (GT), nicotinamide adenine dinucleotide tetrazolium reductase (NADH-TR) and adenosine triphosphatase (ATPase, pH 4.4) [20]. Desmin staining was performed by immunohistochemistry on samples from horses used in the proteomic analysis [21]. Sections were evaluated for the presence of internalized myonuclei, acute necrosis, macrophages, basophilic regenerative fibers, and mitochondrial staining pattern. Central nuclei were scored as; 0 = not present, 1 = present in approximately 10% of fibers in the biopsy, 2 = present in approximately 11–25% of fibers in the sample, 3 = present in more than 25% of fibers in the sample. 

Muscle fiber types were determined using ATPase stains pre-incubated at pH 4.4. Muscle fiber type composition for type 1, 2A, and 2X fibers were determined from counting approximately 250 muscle fibers per horse [22]. 

### 2.4. Analyses of Signalment, Plasma CK, AST and Fiber Composition

Age, timing of biopsy after exercise and fiber type composition were normally distributed (Kolmogorov–Smirnov) and compared between RER-susceptible and controls using an unpaired T test. Plasma CK and AST activities were not normally distributed and therefore log transformed prior to comparison of RER-susceptible and control groups using an unpaired T test. Scores for internalized myonuclei were compared between RER-susceptible and controls using a Mann–Whitney test. GraphPad Prism (version 9.9.9) software (GraphPad, San Diego CA, USA) was used with significance set at *p* < 0.05.

### 2.5. Transcriptomics

#### 2.5.1. RNA Extraction and Sequencing

Total muscle RNA was isolated from snap frozen gluteus medius muscle from all horses as previously described [23]. Sequencing of mRNA (integrity score >7) was performed at the Michigan State University Research and Technology Support Facility Genomics Core (East Lansing MI, USA). Libraries were constructed per horse with a polyA+ capture protocol using the Illumina TruSeq Stranded mRNA Library Preparation Kit (Illumina, San Diego CA, USA) and sequenced on the Illumina HiSeq 4000 platform (2 × 150 base pair) (Illumina, San Diego CA, USA) [23]. Base calling was carried out by Illumina Real Time Analysis (RTA) v2.7.7 (Illumina, San Diego CA, USA) and the output of RTA was demultiplexed and converted to FastQ format with Illumina Bcl2fastq v2.19.1 (Illumina, San Diego CA, USA). 

#### 2.5.2. Mapping and Assembling

Sequence reads were trimmed of adapter sequences (Trimmomatic) [24], filtered of low-quality bases/reads (ConDeTri; Q ≥ 30), and then mapped to the EquCab 3.0 (NCBI reference genome) (HISAT2) [25,26,27]. Sample specific transcriptomes were assembled using StringTie as previously described [25,26]. Alignment statistics and base coverage were obtained with SAMtools [28]. Total gene transcript expression was quantified for unique sequence reads using HTSeq [29]. Gene transcripts with less than two sequence read counts in at least one horse were removed from further analysis to reduce the number of genes with low expression, leaving 13,531 gene transcripts for differential expression (DE) analyses and gene set enrichment. RNA sequence files have been deposited in the NCBI Sequence Read Archive BioProject PRJNA729264, accession numbers SAMN1911538-SAMN19115390 (https://www.ncbi.nlm.nih.gov/sra, accessed 13 October 2022).

#### 2.5.3. RNA-Seq Count Normalization and Transformation

The trimmed mean of M-values (TMM) scaling factors [30] were calculated from the weighted mean of log_2_ expression ratios using the Bioconductor R package edgeR [31]. The TMM normalization factors were used to normalize the library sizes of each sample and remove any systemic technical effects biasing transcript expression between samples. Raw gene transcript abundances were transformed to approximate a normal distribution by calculating the log_2_ counts per million (CPM), which is the log_2_ of the raw counts and scale-normalized library size ratio. The mean-variance trend of gene transcripts was estimated and incorporated in the variance modeling of the DE analysis as precision weights to account for observational level and sample-specific parameters shared across genes [32,33]. The CPM and precision weights were computed using the Bioconductor R package limma [32,33]. Principal component (PC) analysis plots were generated for the CPM gene transcript matrix in R version 3.6.2 (R Foundation for Statistical Computing, Vienna AT) (Appendix A). 

#### 2.5.4. Differential Gene Expression Analysis

Differential expression of gene transcripts (DEG) was identified using limma linear models by weighted least squares [31,34]. Linear combinations of model parameters were used to evaluate the DE between control and RER horses. The fixed effects model with *n* horses and *m* genes can be represented as: y_ij_ = Xβ + *τ*_j_ + ε_ij_; i = 1, …, *n*; j = 1, …, *m*

Where y is a matrix of CPM for every i horse (*n* = 16) and j gene (*m* = 13,531), X is an incidence matrix of fixed effects including overall mean, age, trainer, exercise type, and biopsy time, and β is the vector of covariate estimates. The *τ* vector contains the estimated effects of RER for gene j and the error term as $\varepsilon \sim N\left(0,\sigma_{e}^{2}diag\left(\sqrt{\hat{w}}\right)\right)$ with the estimated precision weights $\hat{w}$ modeling the heterogeneity of the error variance. DE was determined based on *t*-tests relative to a threshold (TREAT) [35] where the threshold used in this analysis was a log_2_ 1.25 [36]. Multiple test correction was performed with false discovery rate (FDR) less than 0.05. 

### 2.6. Proteomics

#### 2.6.1. Sample Preparation and LC/MS/MS

Proteins were extracted from snap frozen gluteus medius muscle samples (5 RER-susceptible and 5 control) in fresh radioimmunoprecipitation assay lysis buffer (Thermo Scientific, Waltham, MA, USA) with protease inhibitor (cOmplete, Mini, EDTA-free, Roche, Basel CH). Sample size was based on the number of samples that could be run on an TMT-11 plex proteomic analysis. Protein concentration was measured by standard BCA assay (Pierce^TM^ Biotechnology, Rockford, IL, USA) and Coomassie-stained SDS gel. LC/MS/MS was performed at the MSU-RTFS Proteomics Core. In brief, 120 ug of protein per samples were subjected to proteolytic digestion with Trypsin/LysC enzyme mix (Promega, Madison, WI, USA) at 1:100 (enzyme:protein) by volume. The resulting peptides were labeled with TMT11-131C (Thermo Scientific, Waltham, MA, USA), one control sample was run in duplicate as an internal control. Tagged peptides were separated and eluted with the Thermo Acclaim PepMap RSLC 0.1 mm × 20 mm C18 trapping column over 125 min at a constant flow rate. Eluted peptides were sprayed into a ThermoScientific Q-Exactive HF-X mass spectrometer (Thermo Scientific, Waltham, MA, USA) using a FlexSpray spray ion source. The top 15 ions in each survey scans (Orbi trap 120,000 resolution at m/z 200) were subjected to higher energy collision induced dissociation with fragment spectra acquired at 45,000 resolution. The resulting MS/MS spectra were processed with Proteome Discoverer v2.2 (Thermo Scientific, Waltham, MA, USA) and searched against the EquCab3.0 UniProt:UP000002281 protein database appended with common laboratory contaminates (cRAP project) using two search engines: Mascot (Matrix Science, London, UK; version Mascot in Proteome Discoverer 2.2.0.388) and X! Tandem (The GPM, thegpm.org; version X! Tandem Alanine 2017.2.1.4), to annotate the peptide spectra. Mass spectrometry proteomic data are available via the ProteomeXchange Consortium PRIDE repository with identifier PXD026121. 

#### 2.6.2. Quantitative Data Analysis

Scaffold Q+ (version Scaffold_4.10.0, Proteome Software Inc., Portland, OR, USA) was used to quantitate TMT-11 plex labeled peptide and to probabilistically validate protein identifications. Peptide identifications were accepted if they could be established at greater than 95.0% probability by the Scaffold Local FDR algorithm. Protein identifications were accepted if they could be established at an FDR ≤ 0.01 and contained at least two identified peptides. A second filter was applied to remove proteins with missing values. Proteins that contained similar peptides and could not be differentiated based on MS/MS analysis alone were grouped to satisfy the principles of parsimony. Proteins sharing significant peptide evidence were grouped into clusters. Spectra data were log-transformed, pruned of those matched to multiple proteins, and weighted by an adaptive intensity weighting algorithm. Of 5867 spectra in the experiment at the given thresholds, 4775 (81%) were included in quantitation. DEP between control and RER-susceptible horses were determined as described in the DEG analysis [32,37]. The model used to identify DEP excluded preceding exercise type and time of biopsy covariates as they did not have a significant effect on protein expression (PCA, Appendix A. The j in this model represents *m* proteins (*m* = 371). Significant DEP was determined using a multiple test correction cutoff of FDR ≤ 0.10.

### 2.7. Transcriptome and Proteome Co-Inertia Analysis

A co-inertia analysis (CIA) was conducted to access the correspondence between the collected transcriptomics and proteomics datasets. CIA is a multi-variate statistical method similar to canonical correlation analysis that aims to quantify the co-variability between two datasets. This correspondence is a global measure of variability known as the “co-intertia” parameter. To estimate this parameter, we used the gene and proteins quantified for the ten horses selected for proteomics. CIA was performed on the centered and scaled log-cpm of genes and proteins with an identity matrix as positive weights for the samples and the Eluclidean metric of log-cpm as positive weights for the gene and proteins, respectively [38]. Pairs of optimal co-inertia loading vectors were estimated via eigenvalue decomposition as described in Min et al. [38]. The first two loading vectors of each omic-dataset were used to select the top 20 divergent genes and proteins from each quadrant and evaluated for pathway enrichment. 

### 2.8. Enrichment and Pathway Analysis

Proteins and genes exhibiting significant differential expression between control and RER horses were analyzed using the R package clusterProfiler for GO, Reactome, and Kyoto Encyclopedia of genes and genomes (KEGG) pathway enrichment analysis [39,40] based on the hypergeometric distribution [36,41]. Gene symbols were converted to ENTREZ gene IDs using the human annotation (org.Hs.eg.db: Genome wide annotation for Human) [42,43]. The reference background used in the enrichment analysis consisted of the list of quantified genes and proteins in the gluteal muscle biopsies with an equivalent human annotation. Pathway enrichment was evaluated separately for the DEP and DEG with negative log_2_ FCs (down-regulated) and positive log_2_ FCs (up-regulated) with their respective background. Two amalgamated enrichment analyses, one for the list of DEG and DEP, and the other for the top divergent genes and proteins from CIA were performed using a combined background. Pathways with significant enrichment relative to background protein/gene expression were determined after multiple test correction with an FDR ≤ 0.05. The enrichment analyses were examined for biological processes, cellular location, molecular function, Reactome, and KEGG pathways [39,40,44]. 

### 2.9. Muscle Biochemistry

Based on the results obtained from the transcriptomic and proteomic data, targeted biochemical analyses were performed to quantify specific oxidants/antioxidants. 

#### 2.9.1. Reactive Oxygen Species (ROS)

Ten ml of protein homogenate was taken from the initial protein isolation and diluted 20× in 1× radioimmunoprecipitation (RIPA) buffer for glutathione and reactive oxygen species (ROS) quantification. An oxidant sensitive fluorescent probe kit (OxiSelect STA-347, Cell BioLabs, San Diego CA, USA) was used to measure total ROS which included reactive nitrogen species, hydrogen peroxide, nitric oxide, peroxyl radicals, and peroxynitrite anions. Sample concentrations were measured using Synergy h1 plate reader (Biotek, Winooski, VT, USA) with 480 nm excitation and 530 nm emission in a 96-well black plate with white wells. All samples and standards were measured in triplicate and concentrations were measured as hydrogen peroxide (H_2_O_2_) equivalence. 

#### 2.9.2. Coenzyme Q_10_

Coenzyme Q_10_ (CoQ_10_) analysis was performed on the resting muscle samples at the Michigan State University Mass Spectrometry and Metabolomics Core using a high-resolution/accurate-mass UHPLC-MS/MS system consisting of a Thermo Vanquish UHPLC interfaced with Thermo Q-Exactive according to Pandey 2018 [45]. Approximately 10 mg of tissue was homogenized in a 95:5 ethanol:2-propanol solution containing 500 ng/mL CoQ4 internal standard with 125 µg of butylated hydroxytoluene pre-dried in the homogenization tube. CoQ_10_ was extracted from this homogenate by adding 400 µL of hexanes then 200 µL milli-Q water. The organic phase was collected, evaporated and then reconstituted in 2 mL of ethanol containing 0.3 M hydrochloric acid. Ten µL of sample was injected onto a Waters Acquity BEH-C18 UPLC column (2.1 × 100 mm) and eluted using a 5 min isocratic flow of 5 mM ammonium formate in 2-propanol/methanol (60:40 *v*/*v*) at 0.3 mL/min. Compounds were ionized by electrospray operating in positive ion mode with a spray voltage of 3.5 kV, capillary temperature of 256.25 °C, probe heater temperature of 412.50 °C, and S-Lens RF level of 50. Spectra were acquired using a full MS/AIF (all ion fragmentation) method at 70,000 resolution, AGC target of 1 × 10^6^ and mass range of m/z 150–1000. The normalized collision energy for the AIF scans was set to 22 V. Data were processed using Xcalibur software version 4.3).

#### 2.9.3. Glutathione

LC/MS/MS was used for the determination of total glutathione following extraction from 30 mg muscle homogenized in 400 μL deionized water (pH 7.0) containing N-ethylmaleimide. The protein precipitate was then reduced using Tris (2-carboxyethyl phosphine) and the extraction was repeated. An internal standard of 20 uM GSH ammonium salts D-5 (Toronto Research Chemicals, North York, ON, Canada) was added to all samples and standards. Chromatographic separation of the thiols was achieved using a Waters Acquity UPLC^®^ HSS T3 1.8 μM (2.1 × 100 mm) column (Waters Corp. Milford, MA, USA) with a gradient elution consisting of acetonitrile and deionized water with 0.1% formic acid. Multiple reaction monitoring, optimized using Waters QuanOptimize software, was used for detection of ions generated by glutathione in the MS/MS detector.

#### 2.9.4. Statistical Analyses of ROS, Glutathione and Coenzyme Q_10_

ROS, glutathione and CoQ_10_ concentrations were normally distributed (Kolmogorov–Smirnov) and compared between RER-susceptible and controls using an unpaired t test in GraphPad Prism (version 9.9.9) software (*p* < 0.05).

## 3. Results

### 3.1. Horses and Muscle Histochemistry

There were no significant differences in age, time between exercise and sampling, plasma CK and AST activities between the RER-susceptible and control Standardbred horses (Table 1). There were no histopathologic abnormalities in control horses. Internalized myonuclei were present in mature gluteal muscle fibers in significantly greater numbers of RER-susceptible Standardbreds (median score RER 1.00, range 0–3; median score control 0, range 0, *p* = 0.0016). Internalized myonuclei were present in 8/9 RER-susceptible horses including those that had not had an acute episode of rhabdomyolysis for 8–10 weeks. (Figure 1A, Table 1 and Appendix A). Basophilic fibers with large central nuclei (early regeneration), acutely degenerate fibers, macrophages, and other inflammatory infiltrates were not observed in any of the muscle samples. One RER-susceptible horse had a cluster of smaller fibers with small internalized myonuclei that had the appearance of late-stage regeneration. No aggregates of desmin or regenerative fibers with dark desmin staining were observed in any samples. RER-susceptible horses had significantly more type 1 (*p* = 0.007, mean 6% more) and a trend to fewer type 2X (*p* = 0.08, mean 6% fewer) fibers than control horses (Figure 1B, Table 1 and Appendix A). ATPase staining could not be obtained in one control horse because sections would not remain affixed to the slides during staining (Appendix A). 

### 3.2. Transcriptomics

#### 3.2.1. Expressed Gene Transcripts

On average 46.5 ± 4.0 million short-read pairs were sequenced per sample library. Adapter and quality filtering removed 9.8% of reads. The retained sequence reads were mapped to the EquCab 3.0 reference genome at a 98% efficiency. Only the uniquely mapped reads were used to quantify transcript abundance (88% of total sequenced read pairs). The average depth of coverage per sequenced base was 33X with a 7% coverage of the reference genome. A total of 29,068 gene transcripts (all annotated genes + 849 novel transcripts) were expressed. After filtering for low count transcripts, 13,531 (56.6%) remained for the differential expression analysis. Of those, 13,040 were autosomal genes, 489 were mapped to the X chromosome and 2 were mitochondrial. Thirteen thousand and fifty-six genes were annotated to locus identifiers and four hundred and seventy-five were novel transcripts unannotated in the current reference genome.

**RER-susceptible differential gene expression:** There was a total of 791 DEG out of 13,040 expressed gene transcripts; 358 were downregulated and 433 were upregulated in RER-susceptible versus controls (range: −3.08 to 7.98 log_2_ FC, FDR ≤ 0.05) (Appendix A). The five annotated transcripts with the highest log_2_ FC were sulfiredoxin *SRXN1* (log_2_ FC 7.98) that reduces H_2_O_2_ to H_2_O by oxidizing sulfenic to sulfinic acid, chitinase *CHIT1* (log_2_ FC 6.93) secreted by activated macrophages, *CXCL1* (log_2_ FC 6.79) an inflammatory chemokine ligand, *PRUNE2* (log_2_ FC 6.15) involved in apoptosis and cell transformation, and an antileukoproteinase (LOC111767890, log_2_ FC 6.07) (Table 2 and Figure 2). The five annotated genes that had the lowest log_2_ FC were ENTH domain containing 1 (*ENTHD1,* log_2_ FC −4.06), which enables phospholipid binding, DNA topoisomerase I mitochondrial (*TOP1MT,* log_2_ FC −3.93), which modified DNA topology, G Protein-Coupled receptor 156 (*GPR156*, log_2_ FC −3.55), a cell surface receptor, ubiquitin thioesterase 44 (*USP44* log_2,_ FC −3.38), a deubiquitinating enzyme, and pipecolic acid, and sarcosine oxidase (*PIPOX,* log_2_ FC −3.31) a proteosomal catabolic enzyme (Appendix A and Figure 2).

#### 3.2.2. Gene Ontology Pathway for RER Transcriptomic Analysis

**Biological Process:** Of the 433 up-regulated genes, 385 were used in the enrichment analysis that identified 150 significant GO terms for biological processes in RER-susceptible versus control (Appendix A). GO terms grouped into categories of inflammation, cell proliferation, hypoxia/stress response, protein modification, extracellular matrix, regulatory processes, amino acid metabolism, mitochondria, and other processes (Table 2 and Appendix A and Figure 3). Inflammatory processes upregulated in RER-susceptible horses involved GO terms covering myeloid leukocyte activation, immune response, leukocyte degranulation, innate immune response, regulation of cytokine production, and tumor necrosis factor (Figure 4). Of the 358 down-regulated genes, 305 were used in the enrichment analysis that identified 189 GO terms for biological processes in RER-susceptible versus control horses. GO terms grouped into categories of calcium ion regulation (terms containing *CASQ* or *CACNA2D1),* purine nucleotide metabolism, electron transport, metabolic processes, fat metabolism, amino acid metabolism, the tricarboxylic acid cycle, polysaccharide metabolism, and other processes (Table 2 and Appendix A). Mitochondrial respiratory chain and NADH dehydrogenase complexes were down-regulated in RER-susceptible horses (Figure 3).

**Cellular component:** There was a total of 372 upregulated genes in 23 enriched GO terms identified in cellular components for RER-susceptible horses (Appendix A). GO terms were categorized into groups that included the proteosome (7 GO terms, 4–15 genes), the extracellular matrix (5 GO terms, 5–31 genes), inflammatory cells/platelets (9 GO terms 4–27 genes) (Figure 5), the cell surface (1 GO term, 27 genes) and endoplasmic reticulum (1GO term 16 genes) (Appendix A). There were 297 down-regulated genes in 30 enriched GO terms identified in cellular components for RER-susceptible horses (Appendix A) that were categorized into mitochondria (25 GO terms, 3–73 genes), t tubule/voltage gated calcium complex (2 GO terms 4–6 genes), the Z disc (1 GO term, 9 genes), transporters (1 GO term, 11 genes), and dihydrolipoyl dehydrogenase (1 GO term, 3 genes) (Appendix A). 

**Molecular function:** There were 11 significant GO terms enriched in molecular function in upregulated DEG for RER-susceptible versus control horses relative to background expression (Appendix A) with functions including extracellular matrix, heme binding and antioxidant activity. In RER-susceptible versus control mares there were 55 enriched GO terms grouped into mitochondrial functions (29 GO terms 3–75 genes), metabolic function (9 GO terms, 3–47 genes), ion transporter (14 GO terms, 3–32 genes), and fat metabolism (3 GO terms, 3–4 genes) (Appendix A).

### 3.3. Proteomics

#### 3.3.1. Confident Protein Classification, Differential Expression and GO Analysis

Of the 727 proteins quantified by mass spectrometry, 403 had <0.1% probability of incorrect protein identification and greater than two identified peptides per protein. Filtering for proteins with missing values removed 6.5%, leaving 371 proteins expressed across all horses. The DE analysis identified 72 DEP in RER-susceptible compared to control horses (Appendix A). Of these proteins, 50 had increased and 12 decreased expression in RER versus controls (−0.07 to 1.91 log_2_ fold change (FC); FDR ≤ 0.10; Figure 2 and Table 3 and Table 4). 

The 50 skeletal muscle proteins with increased differential expression included proteins involved in the mitochondria (*N* = 22), sarcomere proteins (*N* = 17), metabolism (*N* = 9), heat shock factors (*N* = 2), inflammation (*N* = 1), mitochondrial antioxidant (*N* = 1), and 7 proteins with a variety of other functional terms (Table 3 and Table 4). The decreased DEP (*N* = 12) included proteins involved in mitochondria (*N* = 2), myoplasmic antioxidants (*N* = 2), inflammation (*N* = 2), myofilaments (*N* = 1), and other functions (Table 3 and Table 4). 

#### 3.3.2. Gene Ontology (GO) Pathway for RER Proteomic Analysis

There were no significant GO terms identified for upregulated proteins in biological processes, metabolic processes or cellular components, or downregulated cellular components for RER-susceptible versus control horses relative to skeletal muscle background protein expression (Appendix A). Neither were any terms significantly enriched when analyzing all 62 annotated DEP. There were overlapping regulatory roles for the 12 downregulated proteins in biological processes that included metabolic processes (15 GO terms, 5–12 proteins), response to stimulus (8 GO terms, 5–8 proteins), protein modification (7 GO terms, 4–12 proteins), phosphorylation/kinase activity (7 GO terms, 3–5 proteins), signal transduction (7 GO terms, 6–7 proteins), inflammation (7 GO terms, 3–10 proteins), cell proliferation (8 GO terms, 5–6 proteins), coagulation (3 GO terms, 4 proteins), and other miscellaneous processes (3 GO terms, 3–6 proteins) (Appendix A and Figure 3). Signaling receptor binding (6 proteins) was the sole term enriched for molecular function in downregulated DEP.

#### 3.3.3. Differentially Expressed Genes with Differentially Expressed Proteins

There were 13 DEG that also had differential protein expression in the proteomic and transcriptomic datasets (Table 3 and Table 4). Eleven localize to mitochondria (NDUFS2, CYC1, ATP5PF, CS, FH, IDH3B, SUCLA2, MRPS36, PPA2, SLC25A4, LOC100053634 (cytochrome b-c1 complex subunit Rieske)) and two to the cytoplasm (PRDX2, GOT1). Except for fumarate hydratase (FH), and the cysteine-based antioxidant peroxiredoxin 2 (PRDX2), these DEP were all upregulated proteins with downregulated genes in RER-susceptible versus controls. For FH and PRDX2, both DEP and DEG were downregulated in RER-susceptible versus controls. 

### 3.4. Reactome for Merged Differentially Expressed Genes and Proteins

Reactome pathway analysis of combined DEG and DEP datasets revealed 135 significantly enriched Reactome terms containing a total of 135 gene IDs (see Appendix A for individual R-HSA terms). R-HSA terms categorized by function included cell proliferation, inflammation and NFkB signaling, cellular/oxidative stress, protein processing, transcription, metabolism, electron transport, the TCA cycle, extracellular matrix, signaling, transporters, and striated muscle contraction (Figure 6, Table 2 and Appendix A). 

### 3.5. Co-Inertia Analysis

The global similarity between transcriptomics and proteomics (RV-coefficient) was 0.675. The cumulative proportion of variance estimated from the first two pairs of loading vectors were 0.914 (0.715 and 0.200, respectively). There were 71 proteins and 76 genes selected as the top divergent variables from the omics sample space. Twenty-five of the DEP and eight of the DEG were among the top selected in the CIA (Appendix A). One hundred and six annotated genes/proteins were used in the GO analysis. The convergent pathways in this analysis were purine/ribonucleotide metabolism (21 GO terms), oxidative stress (15 GO terms), oxidative metabolism (14 GO terms), ion transport (13 GO terms), other metabolism (11 Go terms), electron transport (8 GO terms), sarcomere/contraction (6 GO terms), hormone response (5 GO terms), cell development (3 GO terms), and other processes (10 GO terms) (Appendix A). 

### 3.6. Muscle Biochemistry

ROS and glutathione concentrations did not differ between RER-susceptible and control horses (Figure 7A,B). CoQ_10_ concentrations were significantly higher in RER-susceptible versus control horses (Figure 7C).

## 4. Discussion

In a previous study of RER in French Trotters, upregulation of genes involved in oxidative stress, cellular stress, and inflammation as well as down-regulation of genes involved in calcium regulation and oxidative metabolism were identified within 24 h of acute rhabdomyolysis (*N* = 5 RER horses, CK range 269–72,000 U/l, mean 9183 U/L) in RER compared to control horses [9]. The question arising from this previous study was whether these changes represent alterations resulting from acute rhabdomyolysis or whether they were in fact chronic underlying molecular characteristics of RER. By examining Standardbred horses susceptible to, but not currently experiencing rhabdomyolysis, we confirmed that altered expression of genes involved in pathways of calcium regulation, oxidative metabolism, cellular/oxidative stress, and inflammation persist in gluteal muscle of Standardbred horses weeks after an episode of rhabdomyolysis. In addition, our study revealed enriched pathways of muscle regeneration (cell proliferation) between episodes of rhabdomyolysis. Thus, our study extends and concurs with findings in French Trotters with acute RER, a breed that has overlapping bloodlines with North American Standardbred trotters, and suggests ongoing molecular derangements often exist even between episodes of rhabdomyolysis in susceptible horses.

There were six DEG from Barrey et. al. [9] acute rhabdomyolysis study identified in the present study, including an antioxidant thioredoxin (*TXN*), the oxoglutarate/malate carrier (*SLC25a11*) that shuttles glutathione and malate into mitochondria [46], malate dehydrogenase (*MDH*), a signaling molecule involved in angiogenesis inflammatory response and fatty acid metabolism *CD36*, a vascular endothelial growth factor *VEGF,* and an interstitial collagenous matrix protein lumican (*LUM*). The latter three genes are all components of cellular regeneration. Because the 25K oligonucleotide microarray provided a limited set of genes to evaluate, there are likely many other overlapping genes that were not discernible in the previous rhabdomyolysis study [9]. 

Altered expression of *RYR1* and the sarcoplasmic reticulum calcium ATPase (*ATP2A1*) were evident in the acute rhabdomyolysis study of RER [9]. In the present study, genes that regulate calcium release by the calcium release channel RYR1 and modulate sarcoplasmic reticulum calcium stores also had altered expression in RER-susceptible horses (Figure 4). Downregulation was apparent for (1) the dihydropyridine receptor (*CANA2D1*), which activates RYR1 resulting in sarcoplasmic reticulum calcium release, (2) S100 calcium binding protein A11 (S100A11) which colocalizes with S100A1 and increases RYR1 calcium release, (3) calcium/calmodulin-dependent protein kinase type II subunit alpha (*CAMK2A*) involved in phosphorylation of RYR1, resulting in an increase in RYR1 calcium release in the face of beta adrenergic (epinephrine) stimulation and (4) junctophilin (*JPH2*), which controls gaiting of RYR1 (Figure 4) [47,48,49]. Differentially expressed genes impacting sarcoplasmic reticulum calcium concentrations included stromal interacting molecule (*STIM1*) and calsequestrin (*CASQ1*). *STIM1* regulates store operated calcium entry into the muscle cell when sarcoplasmic reticulum stores are depleted [50]. *CASQ1* is a high affinity calcium storage protein in the sarcoplasmic reticulum that was a ↑DEP and ↓DEG in RER-susceptible Thoroughbred mares and a ↓DEG in RER-susceptible Thoroughbred geldings between episodes of rhabdomyolysis in previous studies [7,51]. Enhanced sarcoplasmic reticulum calcium stores and post-translational modulation of RYR1 calcium release by oxidation (S-nitrosylation) or beta-adrenergic-induced phosphorylation are potential mechanisms that could enhance RYR1 calcium release and trigger an episode of rhabdomyolysis. Excessive myoplasmic calcium release is often the final common pathway leading to rhabdomyolysis in many myopathies [52]. 

In further agreement with the study of acute rhabdomyolysis in French Trotters, numerous mitochondrial genes were downregulated in RER-susceptible horses in our study [9]. In the acute rhabdomyolysis study, this was interpreted to indicate a decrease in mitochondrial protein expression arising from mitochondrial buffering of calcium as was found in Thoroughbreds susceptible to RER [7,9]. Our proteomic analysis, however, found that 89% (17/19) of the DEP mitochondrial proteins were upregulated and the genes encoding 11 of these proteins had significantly downregulated gene expression. The higher mitochondrial protein expression in RER-susceptible compared to control horses in our study could reflect differences in fiber types between RER-susceptible and control horses. Compared to control horses, RER-susceptible horses had 6% more type 1 fibers, which typically have a higher mitochondrial content and a trend to fewer (6%) type 2X fibers with lower mitochondrial content. A higher type 1 and lower type 2X fiber type composition has been described with training of Standardbred trotters [53,54,55]. The higher degree of training response could potentially be related to the RER-susceptible horses being on average one year older than controls, inherent differences in fitness, as well as common recommendations to not provide any days off training for RER-susceptible horses [54,55,56]. A number of proteins involved in mechanosignaling and generation of a training response also had increased expression in RER-susceptible Standardbreds including CSRP3, PDLIM3, MYOZ1, and SYNPO2 [57,58,59]. It is difficult to draw conclusions about the contribution of metabolic and myofibrillar pathways toward RER susceptibility due to potential differences in fitness and fiber type compositions. Pathways of oxidative/cellular stress, inflammation, and regeneration, however, are remarkable in our analysis because they are not normal physiologic responses, and their enrichment was unlikely the result of more type 1 fibers in RER-susceptible horses. 

CoQ_10_ concentrations were analyzed because four enriched GO terms involving CoQ_10_ (ubiquinone) synthesis were identified among downregulated DEG for RER- susceptible versus control horses (Appendix A). CoQ_10_, a potent antioxidant and transporter of electrons, was present in higher concentrations in RER-susceptible Standardbreds. Higher CoQ_10_ concentrations could have been impacted by the higher mitochondrial content of type 1 fibers which were more abundant in RER-susceptible horses. 

Oxidative stress was highlighted in both GO and Reactome pathways as well as the Co-inertia analysis in RER-susceptible horses. Oxidative stress arises when the amount of ROS produced overwhelms cellular antioxidant capacity. ROS are largely formed during exercise by complexes I and III in the mitochondrial electron transport system and subunits of these complexes were DEP (upregulated) and DEG (downregulated) in RER-susceptible horses in our study [60]. ROS can also be generated by other enzymes which were downregulated DEG in our study including pyruvate dehydrogenase (*PDHA1, PDHX, PDK3),* alpha-ketoglutarate dehydrogenase (*OGDH*), and branched-chain alpha-ketoacid dehydrogenase (*BCKDHA)* [60]. ROS generation has been documented in healthy Standardbreds during acute exercise based on protein carbonylation assays [61]. In our study, muscle ROS concentrations were assayed several hours after exercise using a fluorescent probe and differences between RER-susceptible and control horses were not detected. This, however, does not preclude elevated muscle ROS concentrations occurring during the exercise that preceded the biopsy by several hours or during acute rhabdomyolysis. 

ROS are initially reduced to H_2_O_2_ by superoxide dismutase (SOD) and we found increased expression of mitochondrial SOD protein in RER-susceptible horses (Figure 8). Antioxidant proteins that reduce H_2_O_2_ to H_2_O had decreased expression in RER-susceptible horses, including the peroxisomal enzyme catalase (CAT) and the thiol-based antioxidant peroxiredoxin 2 (PRDX2) (Figure 8). In addition, *PRDX2* (−1.88 log_2_ FC) had decreased expression in RER-susceptible horses whereas thiols sulfiredoxin (*SRXN1* 7.98 log_2_ FC), thioredoxin (*TXN* 3.47 log_2_ FC) and thioredoxin reductase (*TXNRD* 2.29 log_2_ FC) as well as glutathione peroxidase (*GPX2* 4.55 log_2_ FC *GPX*) all had increased expression (Figure 8). It appears that RER-susceptible horses have alterations in thiol-based antioxidants proteins and genes that reduce H_2_O_2_ with notable downregulation of both protein and genes encoding peroxiredoxin 2. 

Concentrations of the ubiquitous thiol-based antioxidant glutathione (L-γ-glutamyl-L-cysteinyl-glycine) were measured in our study because its synthesis is limited by the availability of cysteine and there was altered expression of cysteine-based antioxidants in RER-susceptible horses. GSH concentrations did not differ between RER-susceptible and control horses. There were, however, several differences in specific DEP and DEG in RER-susceptible horses that would appear to enhance mitochondrial glutathione synthesis from cysteine, glutamate, and glycine (Figure 8) and potentially thus maintain normal muscle glutathione concentrations. This included differential expression of cysteine transporter *SLC1A4,* glutamate transporter *SLC25A12,* and a transporter that shuttles glutathione and malate into the mitochondria *SLC25A11* [46]. *SLC25A11* was also a DEG in the acute rhabdomyolysis study [9]. In addition, genes involved in the synthesis of glutathione from substrates glutamate (*GCLC*, *GCLMA*) and methionine (*CTH*, *AHCYL2*) were upregulated in RER-susceptible horses (Figure 8). Glutamate is synthesized from alpha-ketoglutarate and numerous TCA enzymes that expand the pool of substrates for alpha ketoglutarate synthesis had increased protein and altered gene expression RER-susceptible horses (Figure 8) [62]. 

In total, our results indicate that between episodes of rhabdomyolysis, there is ongoing molecular evidence of an oxidative stress response exemplified by the differential expression of genes and proteins involved in the production and maintenance of antioxidants with notable alterations in protein and gene expression for thiol-based peroxiredoxins, thioredoxins and precursors for glutathione synthesis. Neither this study nor a previous study of acute rhabdomyolysis provides evidence that oxidative stress directly causes rhabdomyolysis in Standardbred horses [9]. Oxidative stress, however, is capable of altering signaling mechanisms and producing post-translational modification of proteins such as RYR1 that could contribute to the pathophysiology of RER [47]. 

Nuclear factor of kappa light chain enhancer of activated B cells (NF-κB), a redox sensitive upregulated family of transcription factors (DEG *NFKB1E*, *NFKB2*) was highlighted in the Reactome analysis in RER-susceptible horses. NF-kB is involved in regulation of inflammation, cell adhesion, migration, proliferation, extracellular matrix remodeling, and cell survival processes, all of which had enriched GO terms in RER susceptible horses (Appendix A) [63]. Upregulated DEG relating to inflammation were found in 62 different GO terms involving the innate immune response, macrophages, lymphocytes, neutrophils, antiviral responses, and cell-to-cell interactions (Figure 5). One of the highest DEG was chitinase (*CHIT1* log_2_ FC 6.93), which encodes a protein secreted by activated macrophages that plays a role in innate and acquired immunity [64]. Two proteins associated with the production of proinflammatory cytokines were also differentially expressed, reticulon-4 (RTN4, upregulated), and macrophage migration inhibitory factor (MIF downregulated) [65,66]. In addition, genes related to cellular stress were upregulated in RER-susceptible horses, including adenosine monophosphate deaminase (AMPD 5.52 log_2_ FC), activated under conditions of hypoxia or cell stress, heat shock proteins HSPA8 and HSPAIA, and genes *HSPB3* (2.73 log_2_FC) and *HSPB1* (3.97 log_2_FC). Thus, even when the last reported episode of rhabdomyolysis was on average 6 weeks prior to biopsy, when serum CK activities (155–963 U/l) were relatively normal and when degeneration, macrophages, and inflammatory cells were inapparent in histologic sections, the muscle of RER-susceptible horses had molecular signatures of NF-κB activation, inflammation, and cellular stress. 

Macrophages make a transitory appearance within the first week of rhabdomyolysis to remove degenerating proteins and organelles [67]. Activated satellite cells then migrate to the region and fuse to form myotubes that have large centrally placed myonuclei [68,69]. Within 3–4 weeks, numerous myofibrils are generated that re-established myofiber sizes and the internally located nuclei squeeze between myofibrils drawn by microtubules to take a subsarcolemmal position [69,70]. Four weeks after muscle degeneration there is usually no evidence of previous damage if there is no chronic underlying myopathy [69,70]. None of the muscle samples in our study had the histologic appearance of early phases of regeneration; however, small, internalized nuclei in normal shaped and sized muscle fibers was apparent in eight of nine RER-susceptible horses. Notably, the highest scores for internalized myonuclei were found in horses that had not had a reported episode of rhabdomyolysis for 6–10 weeks (Appendix A). This agrees with previous reports of internalized myonuclei in mature muscle fibers of RER-susceptible Thoroughbreds [71]. Two DEG that produce beta tubulin (*TUBB2A* 4.14 log_2_ FC and *TUBB6* 5.87 log_2_ FC) were upregulated in RER-susceptible horses and tubulin plays a key role in positioning of myonuclei during regeneration [72]. Thus, our histologic, transcriptomic and proteomic analyses all suggest that there is a chronic regenerative response in muscle of Standardbred horses many weeks following acute rhabdomyolysis. 

The inability to completely standardize the horse’s age, training, diet, exercise, and timing of muscle biopsies between RER-susceptible and control groups was a limitation of our study. The necessity to do field studies of myopathies that are triggered in race-training environments makes complete standardization difficult. Trainers would not permit muscle biopsies to be obtained before exercise and the precise time of biopsy after exercise could not be standardized amongst all trainers. Therefore, the statistical model employed for transcriptomics included as fixed effects age, trainer, exercise type, and biopsy time. The diets fed, and the type of exercise performed by control horses were all represented in the RER-susceptible group. Nevertheless, some differences in training and diet could have impacted gene expression in the RER-susceptible and control horses. 

There were few DEG that had associated DEP in our study and the global similarity between our transcriptomics and proteomics identified in the CIA analysis was moderate at 68%. This could be in part due to the fact that skeletal muscle homogenates contain an abundance of large sarcomeric proteins that can deter identification of smaller proteins using TMT proteomic methods [73]. The TMT proteomic analysis used in the present study provided a means to examine relative abundance of a protein with reference to controls in relatively small samples. RNA-seq studies often infer that altered gene expression implies a similar direction of change in protein expression [74]. In our study, however, the direction of DEG and DEP expression was often opposite. Other transcriptomics and proteomics studies of myopathies have also found an approximately 50% disagreement in the direction of fold change between transcriptomics and proteomic data [7,75]. For example, in Thoroughbred horses-susceptible to RER, mitochondrial proteins had decreased expression whereas mitochondrial genes had increased expression [7]. Thus, interpretation of gene expression as an indicator of protein expression should be performed cautiously, particularly in disease models and with consideration of the time required for altered gene expression to impact protein content. 

## 5. Conclusions

Our integrative transcriptional and biochemical methodologies provide novel information on gene and protein expression in muscle of Standardbred racehorse susceptible to recurrent bouts of exertional rhabdomyolysis. Gluteal samples were taken between episodes of rhabdomyolysis to define underlying characteristics of this disease. The integrated analysis revealed persistent alterations in expression of genes and proteins involved in calcium ion regulation, cellular/oxidative stress, inflammation, and regeneration as characteristic of gluteal muscle from RER-susceptible Standardbreds between episodes of rhabdomyolysis that Notably, differential gene expression of calcium ion regulation, oxidative metabolism, and cellular/oxidative stress are also characteristics of acute rhabdomyolysis in trotting racehorses which appear to persist for weeks between episodes could therefore potentially serve as therapeutic targets to mitigate rhabdomyolysis and its long-term effects [9].

## Figures and Tables

**Figure 1 genes-13-01853-f001:**
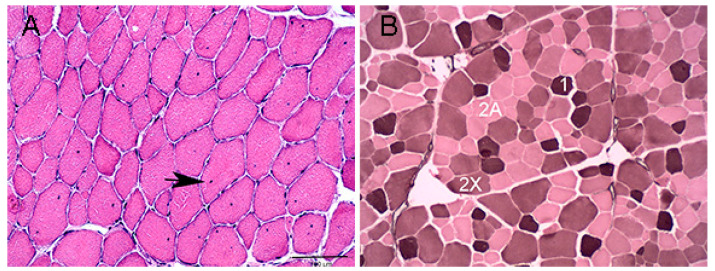
Cross section of gluteal muscle from an RER-susceptible STD. (**A**) Hematoxylin and Eosin stain showing numerous mature myofibers with internalized myonuclei 20×. (**B**) ATPase stain with pH 4.4 acid preincubation showing three muscle fiber types.

**Figure 2 genes-13-01853-f002:**
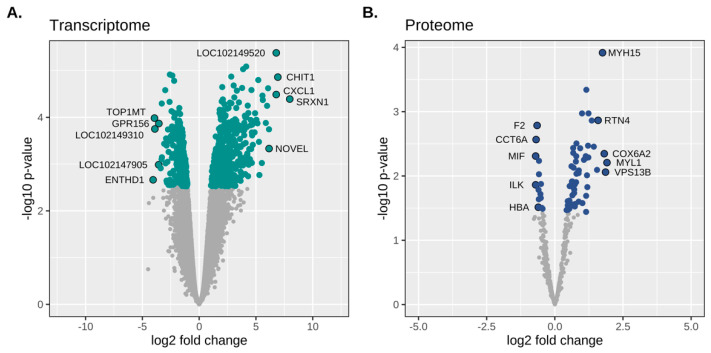
Transcriptomic and proteomic differential expression in RER-susceptible Standardbreds. (**A**). Volcano plot depicting the probability of observing the estimated change in gene expression (−log_10_ scale) on the Y axis and the degree of fold change differences (log_2_ scale) on the X axis. (**B**). Volcano plot depicting estimated *p*-values from differential expression of protein analysis versus log_2_ fold change. There were 791 differentially expressed genes (green dots; FDR ≤ 0.05) and 62 differentially expressed proteins (blue dots; FDR ≤ 0.10) between RER-susceptible and control horses. The genes/proteins with the highest and lowest fold changes are annotated on each graph.

**Figure 3 genes-13-01853-f003:**
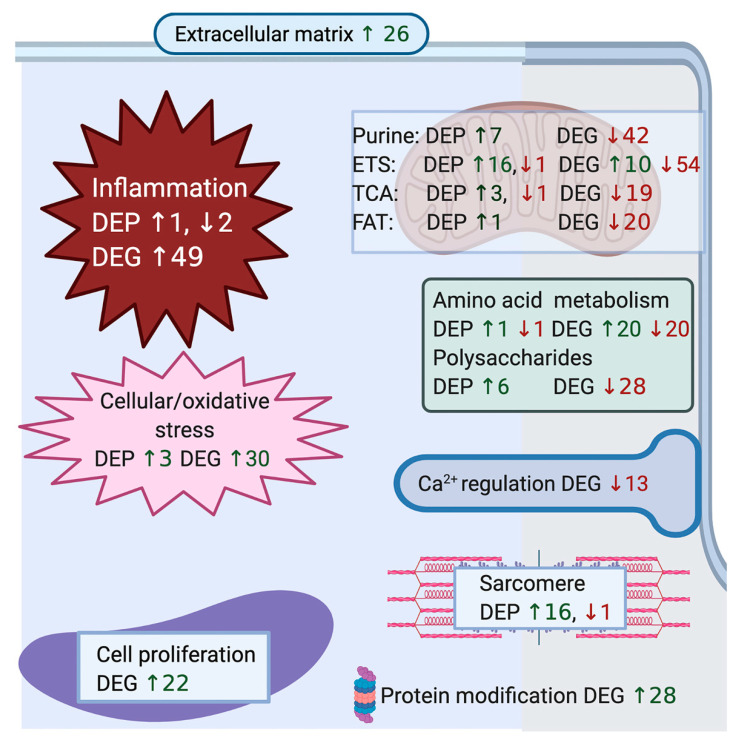
Illustration of convergent pathways for differentially expressed genes and proteins. The number and cellular location of up and downregulated differentially expressed genes (DEG) categorized by convergent pathways for GO terms in biological processes (see Appendix A for specific GO terms and genes) were combined with the number of up and down regulated significantly differentially expressed proteins (DEP) (see Table 3 and Table 4 for specific proteins) in RER-susceptible compared to control horses. The number of genes corresponds to the number of genes in the GO term with the largest gene set for the convergent pathway (Appendix A). DEP and DEG for processes of purine nucleotide metabolism, the electron transfer system (ETS), tricarboxylic acid cycle (TCA) and fat metabolism are depicted within the mitochondrion. Between episodes of rhabdomyolysis, RER-susceptible horses showed upregulation of cellular inflammation, cellular/oxidative stress response, cell regeneration, protein modification and sarcomere proteins. Metabolic pathways were highlighted in the GO analysis in RER-susceptible horses with a notable increase in expression of proteins and decrease in expression of genes related to oxidative metabolism, amino acid, and polysaccharide metabolism. DEG in sarcoplasmic reticulum calcium (CA) transport included downregulation of the dihydropyridine receptor (*CACNA2D1*) and fast twitch calsequestrin (*CASQ1*) and upregulation of slow twitch *CASQ2.* There was no DEP related to calcium regulation in RER-susceptible versus control horses.

**Figure 4 genes-13-01853-f004:**
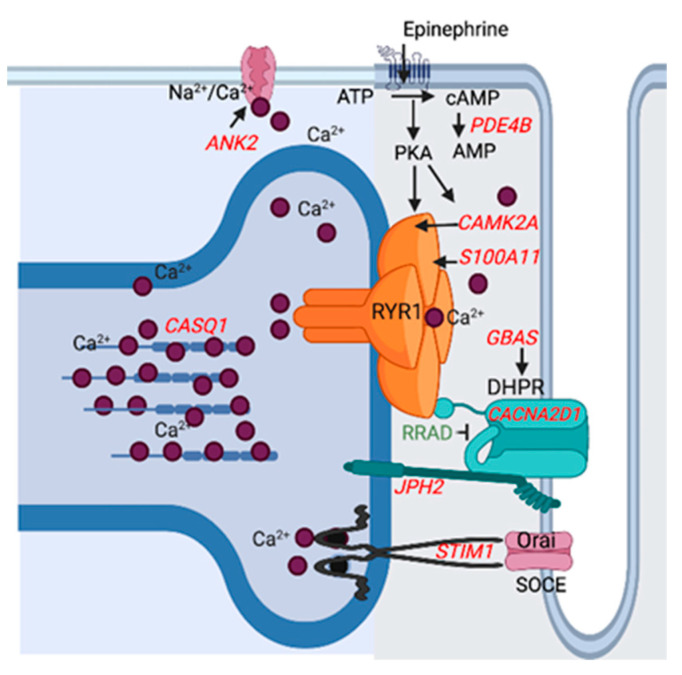
Depiction of the differentially expressed downregulated (red) and upregulated (green) genes that impact sarcoplasmic reticulum calcium stores (*CASQ1, STIM1*) and myoplasmic stores (*ANK2*) or modulate calcium release by the calcium release channel *RYR1*. The dihydropyridine receptor (*CANA2D1*) activates RYR1 resulting in sarcoplasmic reticulum calcium release. Down-regulated Protein NipSnap homolog 2 (GBAS) acts as a positive regulator and upregulated Ras related glycolysis inhibitor and calcium channel regulator (RRAD) an inhibitor of *CANA2D1.* Junctophilin-2 (JP2) inhibits RYR1 calcium release. S100 calcium binding protein A11 (S100A11) colocalizes with S100A1 which increases RYR1 calcium release. Calsequestrin-1 (*CASQ1*) is a high-capacity, moderate affinity, calcium-binding protein that regulates the release of luminal calcium via RYR1. Calcium/calmodulin-dependent protein kinase type II subunit alpha (CAMK2) and protein kinase A (PKA) phosphorylate RYR1, which increases RYR1 calcium release. Beta adrenergic (epinephrine) stimulation activates protein kinase A (PKA) via an increase in adenylyl cyclase which is opposed by phosphodiesterase (*PDE4B*). Stromal interaction molecule 1 (*STIM1*) is a calcium sensor that plays a role in mediating store-operated calcium entry (SOCE), enhancing calcium influx following depletion of sarcoplasmic reticulum stores. Ankyrin 2 (ANK2) is required for coordinate assembly of the Na^2+/^Ca^2+^ exchanger.

**Figure 5 genes-13-01853-f005:**
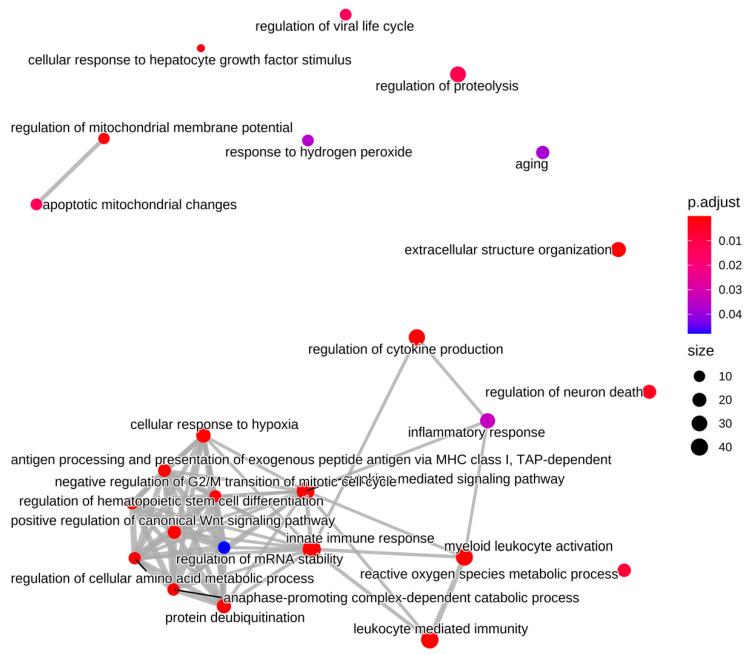
Enrichment map for GO biological processes. Up-regulated differentially expressed genes were enriched for biological processes including inflammation/immune response and reactive oxygen species/antioxidant responses. The size of the vertex indicates the number of DE target genes enriched for that term. The color of the vertex indicates the adjusted *p*-value and the edges (lines) connecting the vertices reflect DE target genes that were common between the GO terms.

**Figure 6 genes-13-01853-f006:**
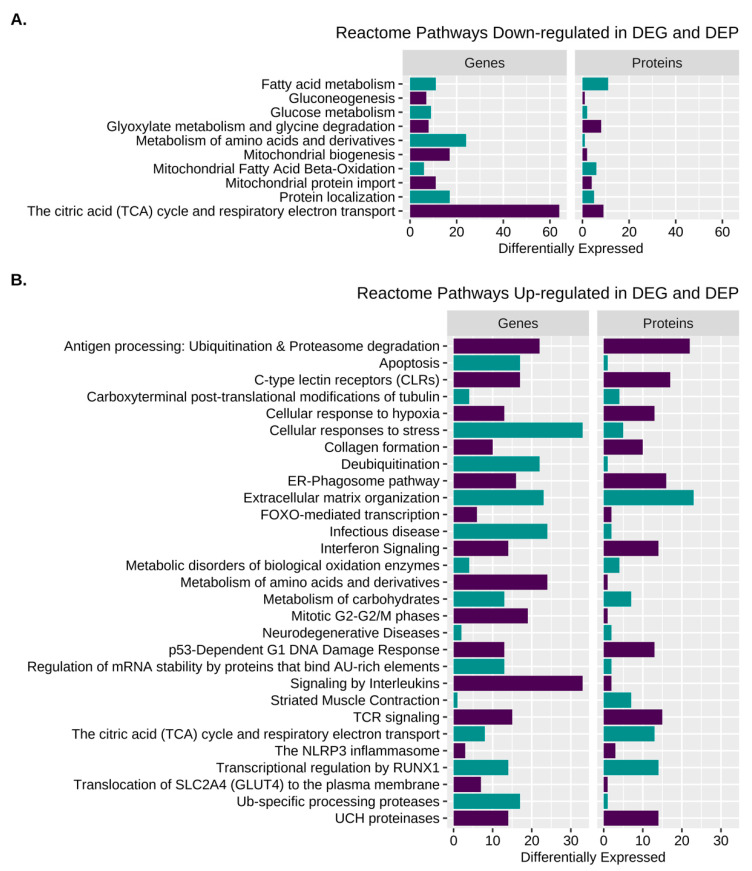
Pathway enrichment for DEG and DEP. (**A**). Enriched Reactome pathways from the amalgamated analysis of B. down-regulated DEP and DEG and (**B**). Up-regulated DEP and DEG.

**Figure 7 genes-13-01853-f007:**
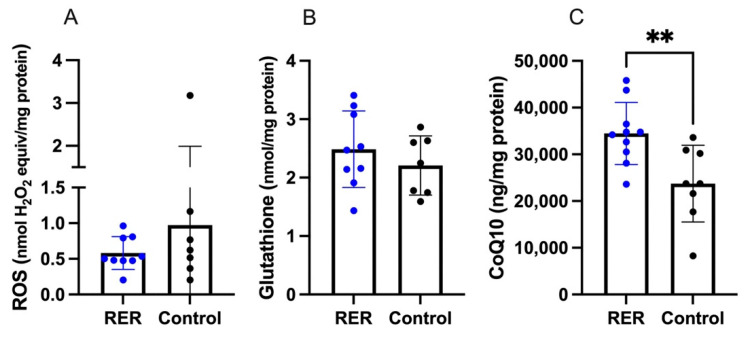
The concentrations of ROS, glutathione and CoenzymeQ_10_ in RER-susceptible compared to control horses. (**A**). ROS concentrations (**B**). Glutathione concentrations. (**C**). CoQ_10_ concentrations. Only CoQ_10_ concentrations were significantly higher (*p* = 0.0072) in RER-susceptible horses. ** indicates *p* < 0.01.

**Figure 8 genes-13-01853-f008:**
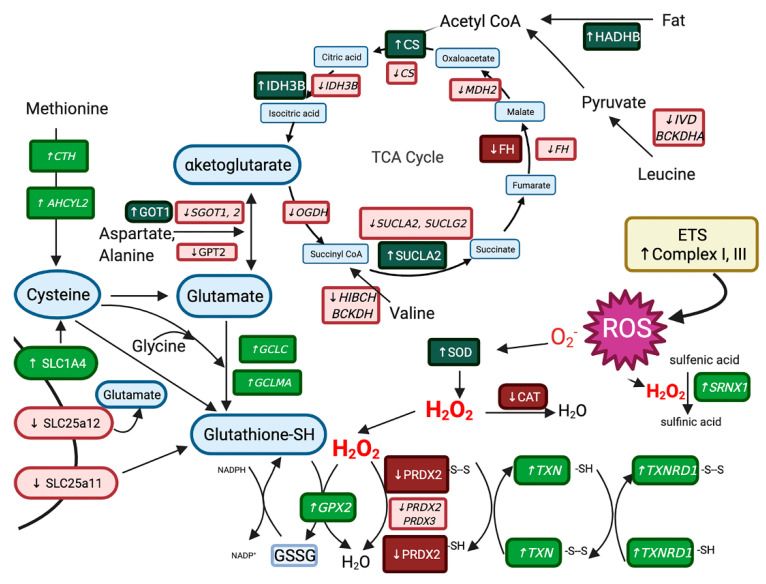
The differentially expressed proteins (dark green upregulated and dark red downregulated) and genes (light green italics upregulated, light red italics downregulated) involving antioxidants, the tricarboxylic acid cycle (TCA), cysteine synthesis, and glutathione synthesis in muscle from RER-susceptible compared to control horses. RER-susceptible horses had increased expression of complexes I and III in the electron transfer system which generated reactive oxygen species (ROS). RER-susceptible horses had increased expression of superoxide dismutase (SOD) which produces H_2_O_2_ and decreased expression of antioxidant proteins catalase (CAT) and the cysteine-based antioxidant peroxiredoxin 2 (PRDX2) as well as increased expression of the cysteine-based genes sulfiredoxin (*SRXN1* 7.98 log_2_FC), thioredoxin, and thioredoxin reductase, which all reduce H_2_O_2_ to H_2_O. Genes involved in the transport and synthesis of cysteine from methionine and the transport and synthesis of the antioxidant glutathione from cysteine, glycine and glutamate were DEG in RER-susceptible versus controls. Proteins (increased) and genes (decreased) in the TCA cycle that produce the glutamate precursor alpha ketoglutarate as well as NADH for the electron transport chain were differentially expressed in RER-susceptible compared to control horses. Abbreviations: *AHCYL2* adenosyl homocysteinase, *BCKDH* branched chain keto acid dehydrogenase, CAT catalase, CS citrate synthase, *CTH* cystathionine gamma-lyase, ETS, electron transport system, HIBCH 3-hydroxyisobutyryl-CoA hydrolase, IDH3 isocitrate dehydrogenase, IVD isovaleryl-CoA dehydrogenase, FH fumarate hydratase, *GCLC* glutamate-cysteine ligase, *GCLM* glutamate-cysteine ligase modifier, *GOT1, GOT2* glutamic-oxaloacetic transaminase, *GPT2*, glutamic-pyruvic transaminase, GSH reduced glutathione, GSSG oxidized glutathione, *GPX2* glutathione peroxidase, *OGDH* oxoglutarate dehydrogenase, *MDH2* malate dehydrogenase, PRDX peroxiredoxin, SOD superoxide dismutase, *SRNX1* sulfiredoxin, SUCL succinate-CoA ligase, *TXH* thioredoxin, *TXNRD1* thioredoxin reductase.

**Table 1 genes-13-01853-t001:** The number of control and RER-susceptible mares, mean (SD) age, sex, time lapse between exercise and muscle biopsy, plasma creatin kinase (CK), and aspartate transaminase (AST) activities and fiber type composition for gluteal muscle type 1, 2A and 2X fibers. Information on individual horses is found in Appendix A. RER-susceptible horses had significantly more type 1 fibers than controls. ^1^ ref. range 127–412, ^2^ ref. range 194–346.

	*N*	Age	Time	CK	AST	Type 1	Type 2A	Type 2X
		Yrs	Hours	U/L ^1^	U/L ^2^	%	%	%
**Control**	7	3.9 ± 1.3	1.8 ± 0.6	257 ± 95	371 ± 96	10.8 ± 1.7	39.3 ± 9.4	49.8 ± 10.0
**RER-susceptible**	9	4.8 ± 1.6	1.7 ± 0.7	433 ± 275	644 ± 564	16.9 ± 4.4	45.3 ± 13.9	37.4 ± 13.9
***p* value**		0.22	0.74	0.14	0.17	0.007	0.37	0.08

**Table 2 genes-13-01853-t002:** Summary of categories for GO terms that were enriched for upregulated and downregulated genes (DEG) in biological processes and pathways in the Reactome (RH) for combined DEG and DEP. The number of GO terms in each category and the range in number of genes within the GO terms in each category are shown. The specific GO terms can be found in Appendix A and Reactome terms in Appendix A.

**Category for GO Term**	**Total Number Genes**	**Number of GO Terms**	**Genes/GO Grouped Category**
Upregulated DEG	385		
Inflammation/Immune response		62	5–49
Cell proliferation		31	5–22
Cell/oxidative stress		31	7–30
Protein modification		7	13–28
Extracellular matrix		7	6–26
Regulatory processes		6	4–23
Amino acid/amine metabolism		5	13–20
Mitochondrial membranes		3	5–10
Ketone metabolism		2	14–16
Response to stimulus		2	19–28
Downregulated DEG	305		
Calcium ion regulation (*CACNA2D1*/C*ASQ*)		39	4–13
Purine nucleotide metabolism		32	5–42
Electron transport		29	4–54
Other metabolism		23	3–75
Amino acid metabolism		15	3–20
Fat metabolism		10	4–20
Carbohydrate metabolism		10	5–28
Tricarboxylic acid cycle		9	4–19
Polysaccharide metabolism		9	5–28
Thermogenesis		5	8–13
Hormone secretion		5	9–16
Other Ion transport		4	4–28
Thioesters		3	9–20
Signaling		3	3–15
Actin		1	5
**Reactome All DEG and DEP**	**Total Number Genes**	**Number of RH Terms**	**Genes/RH Grouped Category**
	498		
Cell proliferation		45	11–18
Immune response		11	15–40
NFkB Immune response		7	14–17
Cell/oxidative stress		8	10–43
Protein processing		8	7–20
Transcription		7	12–19
Metabolism		9	5–47
Electron transport		7	11–48
TCA cycle		5	5–69
Extracellular matrix		5	5–13
Signaling		4	4–14
Transporters		3	12–14
Muscle contraction		2	9–19

**Table 3 genes-13-01853-t003:** Differentially expressed muscle proteins (DEP) in mitochondria of Standardbred horses susceptible to recurrent exertional rhabdomyolysis compared to control. Proteins are identified by their gene ID, protein name, and log_2_ fold change (FC). *p* value adjusted for multiple comparisons (FDR < 0.10) are provided. An asterisk indicates those DEP that also had differentially expressed genes. * indicates DEP that were also differentially expressed genes.

Gene ID	Protein Name	Log_2_FC	Adj. P
**Mitochondria**			
** *Complex I* **			
NDUFB7	Cluster of NADH dehydrogenase [ubiquinone] 1 beta subcomplex subunit 7	1.23	4.75 × 10^−2^
NDUFA2	Cluster of NADH dehydrogenase [ubiquinone] 1 alpha subcomplex subunit 2	0.71	7.20 × 10^−2^
NDUFS2 *	NADH dehydrogenase [ubiquinone] iron-sulfur protein 2	0.47	8.84 × 10^−2^
** *Complex III* **			
UQCRB	cytochrome b-c1 complex subunit 7	1.55	5.20 × 10^−2^
LOC111767815	cytochrome b-c1 complex subunit 8	1.35	4.56 × 10^−2^
LOC100053634	cytochrome b-c1 complex subunit Rieske	0.54	6.19 × 10^−2^
UQCRC1	Cluster of cytochrome b-c1 complex subunit 1	0.64	6.19 × 10^−2^
LOC100051878	Cluster of cytochrome b-c1 complex subunit 6	−0.45	9.21 × 10^−2^
** *Complex IV* **			
LOC100055813	cytochrome c oxidase subunit 6C	1.86	5.20 × 10^−2^
COX6A2	cytochrome c oxidase subunit 6A2	1.81	4.75 × 10^−2^
CYC1 *	cytochrome c1 heme protein	0.67	4.75 × 10^−2^
** *Complex V* **			
ATP5MG	ATP synthase subunit g	1.24	4.56 × 10^−2^
ATP5PF *	ATP synthase-coupling factor 6	1.07	4.75 × 10^−2^
ATP5F1D	ATP synthase subunit delta	0.73	9.00 × 10^−2^
ATP5F1B	ATP synthase subunit beta	0.43	9.39 × 10^−2^
SLC25A4 *	ADP/ATP translocase 1	0.52	8.84 × 10^−2^
** *Fat metabolism* **			
HADHB	trifunctional enzyme subunit beta	1.22	4.75 × 10^−2^
** *TCA cycle* **			
CS *	citrate synthase	0.61	5.20 × 10^−2^
SUCLA2 *	Succinate—CoA ligase subunit beta	0.88	5.20 × 10^−2^
*IDH3B **	isocitrate dehydrogenase [NAD] subunit beta isoform X2	0.77	4.75 × 10^−2^
FH *	fumarate hydratase	−0.51	6.19 × 10^−2^
** *Other mitochondrial* **			
SOD2	superoxide dismutase [Mn] mitochondrial precursor	1.15	4.75 × 10^−2^
VDAC2	voltage-dependent anion-selective channel protein 2	0.85	5.20 × 10^−2^
PPA2 *	Cluster of inorganic pyrophosphatase 2	0.53	9.21 × 10^−2^
MRPS36 *	28S ribosomal protein S36	0.54	8.43 × 10^−2^

**Table 4 genes-13-01853-t004:** Differentially expressed muscle proteins (DEP) located in the myoplasm of Standardbred horses susceptible to recurrent exertional rhabdomyolysis compared to control. Proteins are identified by their gene ID and protein name, log_2_ fold change (FC) and the *p* value adjusted for multiple comparisons are provided (FDR < 0.10). * indicates DEP that were also differentially expressed genes.

	Protein Name	Log_2_FC	Adj. P
**Sarcomere**			
** *Thick and thin filaments* **			
MYL1	myosin light chain 1/3 skeletal muscle isoform isoform X1-fast-twitch	1.91	4.75 × 10^−2^
TNNI1	troponin I slow-twitch	1.43	4.75 × 10^−2^
TPM3	tropomyosin alpha-3 chain isoform X4 slow-twitch	0.93	4.75 × 10^−2^
LOC100062893	myosin 8	0.79	5.20 × 10^−2^
TPM1	tropomyosin alpha-1 chain isoform X5	0.71	6.55 × 10^−2^
MYOM1	Cluster of myomesin-1	0.69	5.20 × 10^−2^
MYL3	myosin light chain 3-slow twitch	0.67	6.19 × 10^−2^
TNNI2	troponin I fast twitch	0.62	8.77 × 10^−2^
MYBPH	myosin-binding protein H	−0.53	7.31 × 10^−2^
** *Z disc* **			
CSRP3	cysteine and glycine-rich protein 3	1.16	4.55 × 10^−2^
FHL1	four and a half LIM domains protein 1 isoform X3	1.13	4.75 × 10^−2^
MYOT	myotilin isoform X1	1.01	8.74 × 10^−2^
PDLIM3	Cluster of PDZ and LIM domain protein 3	1.00	4.56 × 10^−2^
PDLIM5	PDZ and LIM domain protein 5 isoform X9	0.87	5.20 × 10^−2^
DES	desmin	0.79	4.75 × 10^−2^
MYOZ1	myozenin-1 fast-twitch	0.68	4.75 × 10^−2^
SYNPO2	synaptopodin-2	0.63	6.19 × 10^−2^
**Metabolism**			
** *Polysaccharides* **			
AKR1B1	aldose reductase	1.19	6.19 × 10^−2^
ENO2	gamma-enolase	1.18	5.26 × 10^−2^
PGAM2	phosphoglycerate mutase 2	0.75	8.74 × 10^−2^
ALDOC	fructose-bisphosphate aldolase C	0.63	6.19 × 10^−2^
PYGM	myophosphorylase	0.47	8.47 × 10^−2^
GYG	glycogenin 1	0.55	8.74 × 10^−2^
** *Protein* **			
GOT1 *	aspartate aminotransferase cytoplasmic	0.78	6.19 × 10^−2^
PADI2	protein-arginine deiminase type-2	−0.60	6.55 × 10^−2^
** *Lipid* **			
PGP	glycerol-3-phosphate phosphatase	−0.48	9.18 × 10^−2^
**Antioxidant**			
PRDX2 *	Cluster of peroxiredoxin-2	−0.58	5.20 × 10^−2^
CAT	Cluster of catalase	−0.59	8.19 × 10^−2^
**Heat Shock factors**			
HSPA8	heat shock cognate 71 kDa protein	0.79	4.75 × 10^−2^
HSPA1A	heat shock 70kDa protein 1A	0.48	9.39 × 10^−2^
**Inflammation**			
RTN4	reticulon-4 isoform X4	1.59	4.56 × 10^−2^
MIF	macrophage migration inhibitory factor	−0.70	4.75 × 10^−2^
**Other**			
MYH15	myosin-15	1.75	2.41 × 10^−2^
AHSG	alpha-2-HS-glycoprotein	1.15	9.88 × 10^−2^
SPI2	alpha-1-antiproteinase 2 isoform X1	1.15	7.51 × 10^−2^
LOC100065068	alpha-1-antiproteinase 2-like precursor	0.88	8.47 × 10^−2^
EEF1A2	elongation factor 1-alpha 2	0.73	6.19 × 10^−2^
PEBP1	phosphatidylethanolamine-binding protein 1	0.65	7.50 × 10^−2^
YWHAE	Cluster of 14-3-3 protein epsilon	0.58	6.19 × 10^−2^
EHD2	EH domain-containing protein 2	−0.52	7.97 × 10^−2^
APOA1	apolipoprotein A-I	−0.58	4.75 × 10^−2^
HBA	hemoglobin subunit alpha	−0.61	9.11 × 10^−2^
F2	prothrombin	−0.65	4.66 × 10^−2^
CCT6A	T-complex protein 1 subunit zeta	−0.69	4.75 × 10^−2^
ILK	integrin-linked protein kinase isoform X2	−0.70	6.19 × 10^−2^

## Data Availability

RNA sequence files have been deposited in the NCBI Sequence Read Archive BioProject PRJNA729264, accession numbers SAMN1911538-SAMN19115390. Mass spectrometry proteomic data are available via the ProteomeXchange Consortium PRIDE repository with identifier PXD026121.

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
