# Peer review of "Enriched Pathways of Calcium Regulation, Cellular/Oxidative Stress, Inflammation, and Cell Proliferation Characterize Gluteal Muscle of Standardbred Horses between Episodes of Recurrent Exertional Rhabdomyolysis"

_genes, 2022, doi:10.3390/genes13101853_

Round 1

Reviewer 1 Report

The manuscript examines the pathogenesis of RER in Standardbred in detail at the molecular level by analyzing gene and protein expression changes. The manuscript is well written, and the topic is of interest to the field. The materials and methods are clearly described and the conclusions are consistent with the results obtained.

The manuscript described that RER-affected Standardbreds continue to display molecular symptoms over several weeks. Is this molecular symptom associated with recurrence of RER?

In this experiment, individuals with 1-10 weeks after onset were used. Is there a degree of difference in gene and protein expression changes due to post-onset (weeks) differences?

This reviewer was interested in whether the events observed here persist in individuals with RER (forever). Readers may also be interested in these.

Line 309-310: There were "1-10 weeks" difference among the horses with last confirmed RER. Were there any data differences among these RER horses? CV appears to be larger in RER horses (e.g. CK, AST)

Lines 315-317: Does this mean that the RER-STD individuals used do not pathologically exhibit symptoms of RER?

Line 317-319: Isn't the difference in muscle fibers caused by the difference in the INDEL genotype of the MSTN gene?  Did the authors genotype the MSTN gene?

Line 335: Does "29,068 genes" also include splice variants or is it a gene unit?

Line 335-336: "After filtering for low count transcripts"  Analysis may be difficult, but did this exclusion affect the relationship between DEG and DEP?

Line 339: “13,531 genes, 1552 annotations and 475 non-annotations”. This reviewer was confused by this number relationship. How are the remaining genes counted?

Lines 520-524: Do the authors conclude that while pathological RER symptoms were not observed (lines 315-317), RER-susceptible horses continue to be observed as inflammation at the molecular level?

Author Response

The manuscript examines the pathogenesis of RER in Standardbred in detail at the molecular level by analyzing gene and protein expression changes. The manuscript is well written, and the topic is of interest to the field. The materials and methods are clearly described and the conclusions are consistent with the results obtained.

The manuscript described that RER-affected Standardbreds continue to display molecular symptoms over several weeks. Is this molecular symptom associated with recurrence of RER?

In this experiment, individuals with 1-10 weeks after onset were used. Is there a degree of difference in gene and protein expression changes due to post-onset (weeks) differences?

This reviewer was interested in whether the events observed here persist in individuals with RER (forever). Readers may also be interested in these.

Line 309-310: There were "1-10 weeks" difference among the horses with last confirmed RER. Were there any data differences among these RER horses? CV appears to be larger in RER horses (e.g. CK, AST)

Because only 9 RER horses were available to analyze it would be difficult to perform our analysis on horses divided by the number of weeks since their last episode of rhabdomyolysis. We were unfortunately not able to investigate this conclusively.

Lines 315-317: Does this mean that the RER-STD individuals used do not pathologically exhibit symptoms of RER?

The RER-susceptible horses had clinical signs of exertional rhabdomyolysis in the past, but not at the time that we examined them. Because gene expression analysis had previously been performed on horses with acute rhabdomyolysis, we chose to determine what chronic underlying differences might exist in the skeletal muscle of horses that were not currently experiencing rhabdomyolysis but had had episodes in the past. We found that even when RER Standardbreds appeared to have no symptoms, they had alterations in their muscle histopathology that included internalized myonuclei.

We added line 314: Internalized myonuclei were present in 8/9 RER-susceptible horses including those that had not had an acute episode of rhabdomyolysis for 8-10 weeks. (Figure 1A, Table 1, Table S1).

Line 317-319: Isn't the difference in muscle fibers caused by the difference in the INDEL genotype of the MSTN gene?  Did the authors genotype the MSTN gene?

This is true in some breeds of horses, but Standardbred horses do not possess the MSTN variant. Therefore, that would not account for the fiber type differences.

Petersen JL, Valberg SJ, Mickelson JR, McCue ME. Haplotype diversity in the equine myostatin gene with focus on variants associated with race distance propensity and muscle fiber type proportions. Anim Genet. 2014 Dec;45(6):827-35. doi: 10.1111/age.12205. Epub 2014 Aug 26. PMID: 25160752; PMCID: PMC4211974.

Line 335: Does "29,068 genes" also include splice variants or is it a gene unit?

The 29,068 genes correspond to gene units, including annotated genes from the EquCab 3.0 reference genome and 849 novel, unannotated genes discovered in the muscle sample transcriptomes. Genes with overlapping coding regions were combined to a single gene unit while retaining individual gene names.

Line 342 we added A total of 29,068 gene transcripts (all annotated genes + 849 novel transcripts) were expressed.

Line 335-336: "After filtering for low count transcripts" ã€€Analysis may be difficult, but did this exclusion affect the relationship between DEG and DEP?

The main reason for the removal of transcripts with low expression is to reduce sampling noise which can negatively affect the sensitivity of DEG discovery (Sha et. al., 2015). A less conservative threshold of at least 1 transcript count in 70% of the animals per group would add 1,999 genes to our analysis. A DEG analysis for this subset of genes showed no significant difference between group. Out of the 15,537 transcripts with low expression, only 23 (0.15 %) also had its protein quantitated; two of these were DEP (FH and MYH15) with no difference in gene expression. Therefore, considering that a less conservative threshold produced no new DEG and trivial overlap with quantified proteins the filtering threshold used in this analysis has little to no effect on the results obtained for our DEG and DEP analysis.

Reference: https://doi.org/10.1155/2016/3937056

Line 339: “13,531 genes, 1552 annotations and 475 non-annotations”. This reviewer was confused by this number relationship. How are the remaining genes counted?

The number of annotated genes is a typo, the correct number is 13,056 genes annotated in the current reference genome. This error was corrected in the manuscript. Line 345.

Lines 520-524: Do the authors conclude that while pathological RER symptoms were not observed (lines 315-317), RER-susceptible horses continue to be observed as inflammation at the molecular level?

Line 541: We altered this sentence to be clear it is at the molecular level.

By examining Standardbred horses susceptible to, but not currently experiencing rhabdomyolysis, we confirmed that altered expression of genes involved in pathways of calcium regulation, oxidative metabolism, cellular/oxidative stress, and inflammation persist in gluteal muscle of Standardbred horses weeks after an episode of rhabdomyolysis.

Reviewer 2 Report

Valberg et al. present in their study detailed gene and protein expression of gluteal muscle biopsies from Standardbred racehorses who develop recurrent exertional rhabdomyolysis (RER-STD) for unknown reason. N=9 RER-STD and n=7 race trained controls were anaylsed. Muscle biopsies were taken between episodes of rhabdomyolysis. In RER-STD versus controls 791(358 down, 433 up) genes and 62 (50 up, 12 down) proteins were differently expressed. Based on the results targeted biochemical analysis (ROS, Q10, Glutathione) were performed.

The manuscript is well written and the figures and tables are all very illustrative. The study design includes detailed data analysis. The integrative methodology provides novel information about the underlying mechanism in RER-STD. 

Comments:

Histomorphological analysis were performed at muscle biopsies. A Typ-1 fiber predominance and increased number of internalized nuclei were described in RER-STD. It would be interesting to show some histology sections in the manuscript.

Additional, it would be interesting to show a validation of some of the proteomic findings at muscle sections e.g. with immunofluorescence staining.  

Minor comments:

In the abstract and data 3.3. it is described that 50 plus 12 (62) proteins are differentially expressed. In figure 1 in the figure legend 70 proteins (L. 359).  

L. 362-363 is maybe a remnant of the manuscript template

Author Response

Valberg et al. present in their study detailed gene and protein expression of gluteal muscle biopsies from Standardbred racehorses who develop recurrent exertional rhabdomyolysis (RER-STD) for unknown reason. N=9 RER-STD and n=7 race trained controls were anaylsed. Muscle biopsies were taken between episodes of rhabdomyolysis. In RER-STD versus controls 791(358 down, 433 up) genes and 62 (50 up, 12 down) proteins were differently expressed. Based on the results targeted biochemical analysis (ROS, Q10, Glutathione) were performed.

The manuscript is well written and the figures and tables are all very illustrative. The study design includes detailed data analysis. The integrative methodology provides novel information about the underlying mechanism in RER-STD. 

Comments:

Histomorphological analysis were performed at muscle biopsies. A Typ-1 fiber predominance and increased number of internalized nuclei were described in RER-STD. It would be interesting to show some histology sections in the manuscript.

A figure has been added.

Additional, it would be interesting to show a validation of some of the proteomic findings at muscle sections e.g. with immunofluorescence staining.  

We agree that this would be interesting and this could be a good next step in continuing research.

Minor comments:

In the abstract and data 3.3. it is described that 50 plus 12 (62) proteins are differentially expressed. In figure 1 in the figure legend 70 proteins (L. 359).  

Line 367 This had been changed to 62

  1. 362-363 is maybe a remnant of the manuscript template

removed